# Structure and Migration Mechanisms of Small Vacancy Clusters in Cu: A Combined EAM and DFT Study

**DOI:** 10.3390/nano13091464

**Published:** 2023-04-25

**Authors:** Vasileios Fotopoulos, David Mora-Fonz, Manuel Kleinbichler, Rishi Bodlos, Ernst Kozeschnik, Lorenz Romaner, Alexander L. Shluger

**Affiliations:** 1Department of Physics and Astronomy, University College London, Gower Street, London WC1E 6BT, UK; 2KAI—Kompetenzzentrum Automobil- und Industrieelektronik GmbH, Europastrasse 8, 9524 Villach, Austria; 3Materials Center Leoben Forschung GmbH (MCL), Roseggerstraße 12, 8700 Leoben, Austria; 4Institute of Materials Science and Technology, TU Wien, Getreidemarkt 9, 1060 Vienna, Austria; 5Department of Materials Science, Montanuniversität Leoben, Franz-Josef Straße 18, 8700 Leoben, Austria; 6WPI—Advanced Institute for Materials Research (WPI-AIMR), Tohoku University, 2–1-1 Katahira, Aoba-ku, Sendai 980-8577, Japan

**Keywords:** vacancies, diffusion, metals, density functional theory, embedded atom method

## Abstract

Voids in face-centered cubic (fcc) metals are commonly assumed to form via the aggregation of vacancies; however, the mechanisms of vacancy clustering and diffusion are not fully understood. In this study, we use computational modeling to provide a detailed insight into the structures and formation energies of primary vacancy clusters, mechanisms and barriers for their migration in bulk copper, and how these properties are affected at simple grain boundaries. The calculations were carried out using embedded atom method (EAM) potentials and density functional theory (DFT) and employed the site-occupation disorder code (SOD), the activation relaxation technique nouveau (ARTn) and the knowledge led master code (KLMC). We investigate stable structures and migration paths and barriers for clusters of up to six vacancies. The migration of vacancy clusters occurs via hops of individual constituent vacancies with di-vacancies having a significantly smaller migration barrier than mono-vacancies and other clusters. This barrier is further reduced when di-vacancies interact with grain boundaries. This interaction leads to the formation of self-interstitial atoms and introduces significant changes into the boundary structure. Tetra-, penta-, and hexa-vacancy clusters exhibit increasingly complex migration paths and higher barriers than smaller clusters. Finally, a direct comparison with the DFT results shows that EAM can accurately describe the vacancy-induced relaxation effects in the Cu bulk and in grain boundaries. Significant discrepancies between the two methods were found in structures with a higher number of low-coordinated atoms, such as penta-vacancies and di-vacancy absortion by grain boundary. These results will be useful for modeling the mechanisms of diffusion of complex defect structures and provide further insights into the structural evolution of metal films under thermal and mechanical stress.

## 1. Introduction

Copper is a typical face-centered cubic (fcc) metal with a wide range of electronic [1] and electrochemical [2,3] applications due to its reliability, low resistivity, and high thermal and electrical conductivity [1,4]. It is often used at elevated temperatures or in high-stress conditions [4]. Voids are commonly formed in fcc metals under high-stress conditions [5,6] leading to severe degradation effects [7]. They are thought to be caused by the supersaturation of vacancies (V${}_{n}$, with *n* being the number of vacancies) [8,9], which diffuse and aggregate either at grain boundaries or triple junctions [10,11]. For instance, higher stress triaxiality facilitates the formation and growth of micro-voids, which leads to a decrement of strain at failure for rolled Ti${}_{6}$Al${}_{4}$V titanium alloy [12]. Under such conditions, vacancies in fcc metals have been reported to form via two main mechanisms: (i) thermally, after quenching, and (ii) via the nucleation of two-dimensional defects, such as dislocations. In the first case, quenching takes place at temperatures lower than the melting temperature of the metal [13]. In the latter case, migrating dislocations formed via plastic deformation intercept to form steps normal to the slip plane, referred to as jogs. As jogs migrate, rows of vacancies along with lines of interstitial atoms are formed [14,15]. Experimental studies in Al have shown that the clustering of vacancies is already initiated below room temperature [16].

Although methods such as positron annihilation spectroscopy (PES) can be used to investigate vacancies in metals, theoretical simulations are still needed to interpret the experimentally obtained data [17,18]. Therefore, theoretical simulations played an important role in predicting the structural properties of small vacancy clusters in fcc metals [19,20]. Theoretical investigations have shown that di-vacancies in fcc metals are significantly more mobile than mono-vacancies [16,21]. Although di-vacancies appear to play an important role in the aggregation of vacancies in fcc metals [22,23], the main mechanism leading to their high mobility in Cu and inside GBs remains unclear. Moreover, theoretical studies of the properties of vacancy clusters in Al indicate that di-vacancies are unstable [24]. When considering the diffusion of larger clusters, tri-vacancies have been reported to be significantly less mobile compared to di-vacancies, serving mostly as nucleation centers [16]. Penta-vacancies have been regarded as stable clusters, with considerably higher binding energies and migration barriers compared to clusters with less than five vacancies [16,25,26]. For hexa- and higher numbers of vacancies, the first perfect stacking fault tetrahedral (SFT) structures have been reported [27].

The role of vacancies in void formation has been highlighted by theoretical simulations. However, the vacancy clustering mechanisms are not fully understood [16]. In particular, grain boundaries (GBs) can serve as efficient sinks for point defects in metals [28]. Reports have suggested how GBs can reduce the formation and segregation energies of nearby vacancies [29,30]. Based on experimental and theoretical studies, structural alterations and migration of GBs during recrystallization are considered to be the main contributors to the accumulation of vacancies. Thus, vacancy nucleation and agglomeration can lead to the formation of microstructural defects, such as voids, with other studies describing how strain application can lead to the formation of voids at GBs [31]. Despite these insights, the effects of GBs on the diffusion barriers of vacancies remain unclear. In particular, the behavior of di-vacancies inside GBs requires deeper understanding.

One of the main barriers to elucidating these processes is the need for large-scale simulation models. The investigation of different cluster configurations following different migration paths requires a large number of calculations. Therefore, forcefields (FFs) are often employed [16,21], allowing the simulation of thousands or even millions of atoms. However, depending on the number of parameters and fitting process, FFs simulations can lead to results that deviate from those obtained using ab initio approaches [32,33]. Quantum mechanical simulations, on the other hand, are more accurate but are limited by associated computational costs. Thus, optimally, a synergy of the two approaches is required [21,34,35].

In this work, we use both the embedded atom method (EAM) and density functional theory (DFT) to identify the most favorable configurations of small vacancy clusters in Cu and study their migration paths and barriers. By combining both methods, we are able to explain why the di-vacancy diffusion barrier is much lower than that for mono-vacancy in fcc Cu. For larger clusters, we employ the site-occupation disorder code (SOD) [36] and the activation relaxation technique nouveau (ARTn) [37] to accurately and more reliably identify the mechanisms and barriers for their migration. Since voids and vacancies are strongly related to GBs, the interaction range between vacancies and the [100](210) Σ5 twin grain boundary is identified and this GB is shown to act as an efficient sink for vacancies and di-vacancies. The most favorable absorption sites along with the absorption region of mono-vacancies in Cu GB are determined. By combining DFT, EAM, and KLMC [38], we demonstrate that the most favorable configurations of di-vacancies inside the GB introduce strong structural changes. Finally, by comparing the results obtained with DFT and EAM we assess the accuracy of EAM in describing the energetic properties of vacancies in Cu. The results show that, although EAM can accurately describe the defect-induced lattice distortions, significant deviations arise in the main energetic properties of the examined clusters. The discrepancy between the two methods stems from the fact that, unlike DFT, EAM simulations favor the formation of self-interstitial atoms (SIAs) in Cu structures containing lower-coordinated sites, such as larger vacancy clusters and GBs with di-vacancies. These results are useful for understanding the behavior of vacancy clusters in fcc metals and provide further insights into the mechanisms of degradation of metal films under thermal and mechanical stress.

## 2. Methods of Calculations

### 2.1. Computational Details

#### 2.1.1. Embedded Atom Method

The EAM simulations of vacancy clusters in Cu are performed using the Large-scale Atomic/Molecular Massively Parallel Simulator (LAMMPS) code [39]. For the geometry optimization, the conjugate gradient (CG) optimization algorithm is used with a force tolerance of $10^{-10}$ eV/Å. The cells were further equilibrated under an isothermal-isobaric NPT (constant-temperature, constant-pressure) ensemble at constant temperature (0 K) for 0.02 ns using a timestep of 1 fs.

The EAM potential developed by Mishin et al. [40] was used in most calculations. This potential has been fitted semi-empirically and is widely used for pure Cu systems [41]. Even though many reports highlight the efficiency of such potentials, other studies of fcc metals emphasize the need for developing modified EAM potentials for the accurate description of the properties of point defects and the need for further testing [42,43].

#### 2.1.2. DFT Calculations

DFT calculations are performed using the CP2K code [44] with the generalized gradient approximation (GGA) Perdew–Burke–Ernzerhof (PBE) functional [45] and at the Γ point. Due to the large size of computational cells, only one *k*-point is used. CP2K combines the plane wave and Gaussian basis sets (GPW) method [46]. Initially, the cut-off energies for the real-space (RS) integration grid and the mapping of the Gaussians onto the RS grid were converged. An energy cut-off of 550 Ry (7483 eV) is used with a 50 Ry (680 eV) relative cut-off. The Broyden–Fletcher–Goldfarb–Shanno algorithm (BFGS) was used for geometry optimization [47]. The Fermi–Dirac smearing method is used at a 300 K electronic temperature along with the Broyden mixing method, including 10% of the new density [48] to improve convergence. For Cu atoms, an optimized short-range double zeta for valence electron plus polarization functions (DZVP-MOLOPT-SR-GTH) basis set [49] is used with a Goedecker–Teter–Hutter (GTH) PBE pseudopotential [50]. Similar to the previous DFT simulations in Cu [19,51,52], no dispersion corrections have been included in our calculations. These are known to play an important role in calculations of the structure and function of organic/organic and organic/inorganic interfaces [53] and layered systems (e.g., ref. [54]). However, recent studies on the effects of dispersion corrections in description of the properties of bulk metals [55] and metal surfaces [56] are less conclusive and strongly depend on a density functional. Small changes in Cu properties observed in [56] are unlikely to affect our conclusions.

The simulation cell sizes used must be sufficient to include all the lattice distortion effects induced by vacancy clusters in the process of diffusion. To evaluate the simulation cell size needed in order to include these effects and avoid interactions between periodically translated images of the vacancies, different supercell sizes were tested (2 × 2 × 2–6 × 6 × 6). It was found that at least a 4 × 4 × 4 256-atom supercell was needed for clusters with up to 6 vacancies. Therefore, for the NEB calculations of di-vacancies and tri-vacancies a 4 × 4 × 4 256-atom supercell is used. Simulations of clusters with more than three vacancies are carried out in a 5 × 5 × 5 500-atom supercell.

To study the interaction of vacancies with GBs, we choose the [100](210) Σ5 twin boundary, which is one of the lowest energy GBs in Cu [57] and is commonly used in atomistic simulations [52,58]. The simulation cell was periodically translated in the X-, Y- and Z-axes. On the Z-axis, an additional 10 Å vacuum was added in order to avoid interactions between periodically translated images. To simulate the absorption of mono- and di- vacancies in the GB, the 304 (15.66 Å × 5.56 Å × 40.92 Å) and 912 (24.07 Å × 14.35 Å × 40.92 Å) atom cells are used in DFT and EAM, respectively. To compute the diffusion barriers of di-vacancies towards the GB, two simulation cell sizes are used in DFT, 152 and 304-atoms, whereas five sizes are examined using EAM, namely 152, 304, 456, 608, and 912-atom cells. To render the resulting structures, VESTA [59] and OVITO [60] are used. Grain boundary structures are constructed using Atomsk [61].

Calculations of migration barriers of small Cu vacancy clusters (V${}_{1}$, V${}_{2}$, V${}_{3}$) through various diffusion paths are performed using the climbing image nudged elastic band (CI-NEB) method implemented in the CP2K code [62]. For all the investigated diffusion paths, five replicas/images are used. Direct inversion in the iterative subspace (DIIS) is used as the optimizer for the band [63].

Formation energies per added vacancy for different clusters are calculated as:(1)$$E_{\mathrm{form}}=\left(E_{n}-E_{n}-1\right)+\mu_{\mathrm{Cu}},$$
where $E_{n}$ and $E_{n}-1$ are the energies of the crystal with n and n − 1 vacancies. For $\mu_{\mathrm{Cu}}$, the energy of a Cu atom in the bulk is used. Binding energies per added vacancy in a cluster are computed as follows:(2)$$E_{\mathrm{bind}}=\left(E_{n}-E_{n}-1\right)+\mu_{\mathrm{Cu}}-E_{1f},$$
where $E_{1f}$ is the formation energy of a mono-vacancy. Positive binding energy values correspond to the favorable addition of a vacancy.

Vacancy absorption energies at GB are computed as the energy difference between a vacancy located in the bulk and at varying positions in the GB. Negative absorption energy values indicate a favorable position for the vacancy.

### 2.2. Vacancy Cluster Identification

One of the aims of the current study is the identification of the most favorable geometries of vacancy clusters with a different number of vacancies. A cluster is defined as a group of vacancies positioned at a certain distance apart. In order to determine that distance, the interaction range between vacancies needs to be defined. Vacancies have negligible interaction energies (less than 0.05 eV) when separated by more than 5 Å. To reduce the number of configurations of V${}_{n}$ clusters by taking into account the symmetry of the system, the site-occupation disorder code (SOD) is used. The code applies isometric transformations or symmetry operations, such as translations, rotations, and reflections, to determine all unique cluster configurations. If an isometric transformation that can convert one cluster configuration into an already identified one is found, the cluster is not considered further [36]. All nonequivalent configurations identified by SOD are first optimized using LAMMPS/EAM, with the lowest energy configurations being reoptimized using CP2K/DFT.

In the bulk, for V${}_{3}$, V${}_{4}$, V${}_{5}$, and V${}_{6}$ 14, 72, 223, and 870 non-equivalent configurations, respectively, are identified using SOD within a 7.23 Å × 7.23 Å × 7.23 Å cubic 2 × 2 × 2 32-atom supercell of fcc Cu. To determine the structure and energy of each cluster, 500-atom cells containing the cluster are fully relaxed using the EAM potential, and the lowest energy configurations are identified within the range of 10–25 *kT*, depending on the number of vacancies, where *k* is the Boltzmann’s constant and *T* is the room temperature. We consider this energy range sufficient to take into account the thermodynamically accessible structures. The geometries of these structures were further optimized using DFT to identify the lowest energy configurations.

In regions with low symmetry, such as GBs, the knowledge led master code (KLMC) is employed. KLMC is a structure prediction method enabling us to refine input files and generate outputs that can be utilized by third-party electronic structure software [38,64]. A detailed description of how the code operates can be found in ref. [38]. Using the code, one can predict structures not only in the bulk but also at free surfaces. In this study, KLMC is used to identify all possible di-vacancy configurations in the GB. Initially, 1500 V${}_{2}$ configurations generated via KLMC are optimized with EAM. The lowest energy structures within the range of 10 *kT* are reoptimized using DFT.

### 2.3. Calculation of Diffusion Barriers of Vacancy Clusters

The energetically most favorable cluster configurations in the bulk are used as the initial and final configurations for the DFT diffusion barrier calculations of mono-, di- and tri-vacancies using the nudged elastic band CI-NEB method [62]. For such clusters, due to the crystal symmetry, CI-NEB is sufficient to identify all the different diffusion paths. The path with the lowest diffusion barrier is defined as the most favorable.

For larger clusters, the saddle points cannot be identified based on crystal symmetry. To identify and compute the migration barriers of V${}_{4}$, and V${}_{5}$ clusters, the activation relaxation technique nouveau [37,65] is used in tandem with EAM and DFT. In ARTn, nearby local minima are searched by randomly displacing atoms within a certain radius (deformation radius) by a predetermined distance. To identify the cluster migration paths along with the transition states, a three-step process is followed. At the start of the activation, atoms within the deformation radius with respect to the atom closest to the center of mass of the vacancy cluster of interest are displaced along random directions. To achieve this, a nonzero term in the force is created in order for the configuration to escape the harmonic well. The configuration is then shifted by iteratively applying a redefined force to a neighboring saddle point and converged to a new local minimum [66]. For the new local minimum point to be accepted, the energy difference between the two local minima needs to be lower than a certain energy threshold. The same process is repeated for the identification of more saddle points [65,66] until the system reaches an equivalent cluster configuration with the center of mass of the cluster displaced compared to the initial configuration. This three-step process is repeated several times to identify different migration paths. A migration path typically includes several individual vacancy moves and corresponding barriers. Once all the paths are identified, we define the most favorable path as the one with the lowest barriers.

More specifically, each search starts from the lowest energy configuration of each cluster. When the first saddle point is found, the structure is fully optimized using EAM. After the optimization, a second local minimum point is identified using the ARTn code. The search is repeated until the nearest equivalent cluster configuration is found. All the minima and saddle point configurations identified via ARTn/EAM for the lowest energy path are reoptimized using DFT. For the saddle point DFT optimization, the first nearest neighbor atoms from the introduced vacancies are fixed. In the case of V${}_{6}$, since more than 20 configurations are identified using ARTn, only EAM was used for their optimization. To define the accuracy of the saddle point identification process, the deformation radius is chosen at 5 Å. A criterion to accept or reject new minimum points, also referred to as events, is chosen at 0.5 eV. The maximum distance for breaking the crystal symmetry by random displacement of atoms within the deformation radius is set at 0.05 Å.

For V${}_{4}$, five runs are conducted, whereas for larger clusters (V${}_{5}$, V${}_{6}$) the number of runs is increased to ten. For each run, a maximum number of one-hundred events are generated, with acceptance event rates ranging between 91 and 93%. Following this process allows us to understand the initial steps for the deformation and migration of vacancy clusters and preferable migration paths.

## 3. Results of Calculations

The same color code is used in all the figures below, with blue atoms corresponding to Cu atoms, vacancies shown in red, and SIAs depicted in white.

### 3.1. Structure of Vacancy Cluster

Figure 1a,b illustrate the relative and formation/binding energies, respectively, of the lowest energy configuration of V${}_{2}$–V${}_{6}$ clusters identified using the SOD code and optimized with DFT. For the formation and binding energies in Figure 1b, the EAM results are also included. Figure 1c shows the lowest energy configurations of V${}_{3}$–V${}_{6}$ clusters optimized using DFT. The configurations are shown in the same energetic order as in Figure 1a. As expected, vacancies tend to cluster in closed-packed structures. EAM predicted the same lowest-energy tri-vacancy configuration as DFT, which forms the closest packed cluster with all three vacancies relaxing at first nearest neighbor (1NN) distances from each other, as shown in Figure 1c V${}_{3}$ 5. The difference in energies between the lowest and less compact second-lowest tri-vacancy clusters is approximately 0.1 eV.

EAM and DFT identified the same lowest energy configuration for a tetra-vacancy, which is the close-packed cluster shown in Figure 1c V${}_{4}$ 5. The energy difference between the two lowest energy V${}_{4}$ structures seen in Figure 1a optimized with DFT is less than 0.1 eV. These results indicate that rapid transitions between different configurations of V${}_{3}$ and V${}_{4}$ clusters should be taking place even at relatively low temperatures.

The lowest energy configuration of the V${}_{5}$ cluster identified using DFT is shown in Figure 1c, where V${}_{5}$ 5 forms the closest packed penta-vacancy. We note that this is the only case where EAM predicted a different lowest energy configuration compared to DFT. According to EAM, the lowest energy penta-vacancy cluster consists of six vacancies and an SIA located in the geometric center of the cluster, as shown in Figure 1c V${}_{5}$ 1. A similar structure was previously reported in theoretical studies of Al and Cu using other forcefields [16]. Our EAM calculations predict that this structure is lower in energy compared to the second-lowest energy structure by 0.13 eV. However, according to DFT, this configuration is 0.67 eV higher in energy than the pyramidal V${}_{5}$ 5 configuration.

Both EAM and DFT predicted the same lowest energy V${}_{6}$ cluster configuration. As seen in Figure 1c V${}_{6}$ 5, this configuration forms a symmetric octahedral void, the same as the V${}_{5}$ configuration shown in Figure 1c V${}_{5}$ 1 but without the SIA. Both the V${}_{5}$ and V${}_{6}$ configurations shown in Figure 1c V${}_{5}$ 1 and V${}_{6}$ 4, respectively, are the only ones including SIAs predicted by DFT. In the case of V${}_{6}$, this cluster using DFT is 0.14 eV higher in energy compared to the lowest V${}_{6}$ configuration. It consists of three SIAs and nine vacancies. Similar lowest energy V${}_{6}$ configurations have been reported in previous DFT studies [19]. These results indicate that the symmetry of the cluster plays an important role in the stabilization of SIAs.

We note that previous studies that employed forcefields have emphasized the role of SIAs in the microstructural evolution of bcc metals, such as Fe and W, due to their high mobility [67,68]. Our results demonstrate that EAM more readily predicts the formation of SIAs than DFT. A number of stable configurations of V${}_{4}$, V${}_{5}$, and V${}_{6}$ clusters including SIAs obtained with EAM were found to be unstable with DFT. This suggests that EAM potentials are not always accurate for predicting the behavior of SIAs.

### 3.2. Formation and Binding Energies of Vacancies

Previous MD studies [69] illustrated that, as jogs nucleate and dislocations glide, the formed rows of vacancies also include discontinuities. The formation of a vacancy cluster along the path that jogs migrate may affect the distributions of individual vacancies. Thus, it is important to determine the range and strength of the interaction between mono-vacancies and vacancy clusters.

The DFT and EAM results, summarized in Figure 1b, show similar trends for binding and formation energies per added vacancy. These are defined by Equations (1) and (2) above. The plot includes the binding/formation energies of the lowest energy configurations for each cluster obtained through DFT, as seen in Figure 1c. As the number of added vacancies increases, the binding and formation energies per vacancy increase and decrease, respectively. The penta-vacancy formation energy is slightly higher per vacancy than that for the tetra-vacancy due to the less compact configuration. The addition of the sixth vacancy creates a compact and symmetric configuration, significantly reducing the formation energy per vacancy. We need to point out that our DFT and EAM results predict the formation of a di-vacancy as energetically favorable, unlike previous studies in fcc metals [24]. These results indicate that, as the number of attached vacancies in small clusters increases, the addition of the following vacancy becomes more favorable than the decomposition of the cluster. It is still not clear at what number of vacancies the addition of further vacancies would start being unfavorable. However, such an investigation is beyond the scope of the current study.

### 3.3. Cluster Migration Paths in the Bulk Cu

#### 3.3.1. Migration of V${}_{1}$–V${}_{3}$ Clusters

Both EAM and DFT calculations demonstrate that individual mono-vacancies have a very weak (lower than 0.1 eV) interaction with other vacancies and small clusters V${}_{1}$–V${}_{5}$ at distances between the centers of symmetry of the clusters and the mono-vacancy exceeding 5 Å. The most significant interaction takes place at distances below 3.6 Å, indicating that these interactions are very short-ranged. Hence, the diffusion of clusters and vacancies is the only way for them to meet in a perfect lattice to form bigger clusters and voids. To investigate the migration mechanisms of small clusters, different migration paths have been calculated for each cluster using NEB, but only the lowest energy barriers are included in Figure 2a.

The mono-vacancy prefers to hop between the 1NN sites with a barrier of 0.65 eV, in good agreement with previous computational studies reporting barriers of 0.69 eV [70]. More paths are available for a di-vacancy. The paths that required the dissociation of the V${}_{2}$ cluster into two individual mono-vacancies at a next nearest neighbor (NNN) distance are found to have a barrier of 0.55 eV or above. However, when one of the mono-vacancies hops to an adjacent site remaining at the 1NN distance from the other vacancy, the barrier is much smaller at 0.4 eV. The main reason for this reduction is not only due to the non-dissociation of the cluster but also to the reduced crystal distortion energy. When an individual vacancy migrates, an adjacent Cu atom occupies the vacancy site (Figure 2b). This atom has to diffuse through a gate formed by four Cu atoms for such a move to take place. During this process, each of these four Cu atoms is displaced by 0.2 Å. When the two vacancies remain at the 1NN distance during the migration, the number of atoms that need to be displaced is reduced by one. Thus, the energy barrier for cluster diffusion is reduced. This explains why di-vacancies in fcc metals are considered to be more mobile than other clusters.

The migration mechanism for V${}_{3}$ shown in Figure 2c involves two steps: First, one of the vacancies hops into a nearby site, forming a less compact transient configuration of three vacancies. Second, another vacancy hops to restore the compact rectangular configuration at a new position of the center of mass (see Figure 2c). This mechanism involves only single vacancy hops and has a barrier of 0.52 eV.

#### 3.3.2. Elementary Migration Paths of V${}_{4}$–V${}_{6}$ Clusters

In small clusters, the single vacancy hop mechanisms resulted in lower barriers compared to a collective motion of the cluster. In this section, we consider whether this mechanism also works for larger clusters. Since the diffusion of larger clusters involves a substantially larger number of steps, the ARTn code is used to search for adjacent local minima and the corresponding energy barriers. The search continues until a migration path for transitioning from the minimum energy structure found using SOD to the same structure at a different nearby site in the crystal is identified. As the acceptance criterion for new local minima in ARTn we used 0.5 eV and only single vacancy hopping mechanisms were identified.

Barriers identified for migration of the V${}_{4}$ cluster and some of the intermediate configurations are shown in Figure 3a,(bi), respectively. This mechanism involves consecutive jumps of single vacancies, with a DFT calculated barrier of 0.84 eV. As can be seen in Figure 3, this involves a three-step process between two almost equivalent minima which proceeds through two transient configurations and three saddle points. The transient configurations during the migration can be seen in Figure 3b. V${}_{4}$ migration involves the same diffusion mechanism as the one reported for V${}_{2}$ (see Figure 2b). The computed barriers are close to the results of previous studies for fcc Ni, where the 0.66 eV barrier was calculated for tetra-vacancies using EAM [21].

Figure 3 also shows the migration path and barriers for the lowest energy V${}_{5}$ cluster. For identifying the diffusion path and prior to optimizing with DFT, the ARTn search employed the EAM potential developed by Ackland et al. [71]. This potential is used since it predicted the same lowest energy penta-vacancy configuration as DFT (see Figure 1c V${}_{5}$ 5). The highest barrier for this cluster is approximately 0.84 eV. Similarly high barriers were found for V${}_{5}$ in Al [25] using EAM, namely 1.22 eV. Therefore, such clusters will be mobile only at high temperatures. The migration process involved nine configurations: two global minimum points, four local minimum points, and four saddle points. Each step involved the migration of one vacancy (see Figure 3(bii)).

The migration path for the V${}_{6}$ cluster is more complicated. Using ARTn and EAM, we have identified 18 transition configurations with the highest barrier of 0.96 eV. The calculated barriers for all clusters are summarized in Table 1. The highest barrier for the most preferred path is regarded as the diffusion barrier of the cluster. They indicate that the migration of V${}_{4}$, V${}_{5}$, and V${}_{6}$ clusters may become important only at high temperatures reaching 1000 K, as has recently been demonstrated for Ni [72].

#### 3.3.3. Crowdion Motion of Vacancy Clusters

Previous studies have mentioned the collective one-dimensional (1D) crowdion motion as a favorable diffusion mechanism for vacancy clusters in metals, especially in bcc lattices [73,74]. Our DFT results show that the barrier for crowdion motion of a di-vacancy toward the Cu(111) surface is higher by 0.16 eV than the identified lowest barrier of a V${}_{2}$. The most favorable 1D configuration of a V${}_{3}$ cluster is higher in energy compared to the lowest energy V${}_{3}$ configuration by approximately 0.2 eV and 0.15 eV with DFT and EAM, respectively. The crowdion displacement barrier for V${}_{3}$ cluster is by 0.15 eV higher than the most favorable V${}_{3}$ barrier.

EAM also showed that linear crowdion configurations of V${}_{4}$, V${}_{5}$, and V${}_{6}$ clusters were higher in energy by 0.46 eV, 0.78 eV, and 1.22 eV, respectively, compared to the lowest energy V${}_{4}$, V${}_{5}$, and V${}_{6}$ clusters. The same energetic differences using DFT were 0.2, 0.55, and 0.92 eV for V${}_{4}$, V${}_{5}$, and V${}_{6}$, respectively. These results indicate that within both EAM and DFT, V${}_{4}$, V${}_{5}$, and V${}_{6}$ prefer to form close-packed clusters. Therefore, for clusters with a small number of vacancies, a 1D crowdion diffusion would be unfavorable. However, since a series of vacancies and SIAs are expected to form as jog-connected dislocations migrate, our results do not rule out the possibility of crowdion motion of a 1D linear configuration of clusters of more than six vacancies.

## 4. Vacancy Interaction with Grain Boundaries

Our calculations of vacancy migration paths in the bulk Cu have demonstrated that the migration of larger clusters (V${}_{4}$–V${}_{6}$) will require substantially higher energy barriers compared to smaller clusters. However, under thermodynamic equilibrium conditions, such clusters are unlikely to form in perfect Cu due to the high vacancy formation energies [75] even at high temperatures [34]. We investigate how the properties of vacancies are affected under non-equilibrium conditions where 2D defects are present. In particular, vacancies and vacancy clusters can diffuse to GBs and become absorbed by them with a varying degree of efficiency.

### 4.1. Interaction of Mono-Vacancy with Grain Boundary

Figure 4(ai) shows the Σ5 twin grain boundary used to determine how the presence of two-dimensional defects that act as efficient sinks can affect the mobility of vacancies. By introducing a vacancy in each of the numbered sites, the most favorable absorption sites for a vacancy in the GB are identified. The selection of sites was based on previous DFT segregation studies in Cu GB [76,77]. The simulation cells for DFT and EAM comprised of 304 and 912 atoms, respectively, are shown in Figure 4(aii). EAM and DFT results are included in the graph shown in Figure 4b. The plot includes the computed absorption energies both prior to (single point) and after the geometry relaxation at various sites in the GB. Prior to relaxation, EAM and DFT are in good agreement in identifying the equivalent sites 5 and 7 as the most favorable positions, whereas site 6 is found to be unfavorable. The same most favorable segregation sites have been identified in ref. [52] using DFT. We note that our 304-atom cells are considerably larger compared to 76–112-atom cells used in previous similar DFT studies [51,78,79], which allows us to carry out full relaxation of the GB structure as a result of the absorption of vacancies. After the relaxation, both methods show that the vacancy migration from sites 3/9, 4/8 to site 5/7 is accompanied by an energy gain of around 1.0 eV. DFT predicted a stronger GB–vacancy interaction by approximately 0.1 eV compared to EAM.

The interaction range between vacancies and GBs (absorption region) is demonstrated in Figure 4c, where the absorption energies of a mono-vacancy at different distances from the center of symmetry of the GB are calculated. The absorption region is approximately 6.9 Å wide, corresponding to an interaction radius of 3.45 Å between vacancies and GBs. The similar absorption region of 6.36 Å was identified in ref. [80] for mono-vacancies in Σ5 grain boundary in bcc W using EAM. DFT and EAM absorption energies are in good agreement, with both methods predicting a gain of 1.0 eV for the absorption of a mono-vacancy. These results are broadly in agreement with the calculations presented in ref. [81].

### 4.2. Interaction of di-Vacancy with Grain Boundary

Our results, as well as previous data [21], predict the higher mobility of di-vacancies than mono-vacancies in the bulk of Cu. Hence, it is of interest to study the interaction of di-vacancies with the regions that are prone to absorb point defects, such as GBs. First, we search for favorable absorption sites for di-vacancies within 6.9 Å from the GB. Different configurations were tried using a similar process to the one used for the vacancy cluster investigation in the bulk. Due to the larger number of lower-coordinated atoms in GB compared to the bulk, KLMC was used to identify all different configurations of di-vacancies. All identified V${}_{2}$ configurations were optimized with EAM, with the lowest energy structures being reoptimized with DFT.

Figure 5a shows the lowest energy initial V${}_{2}$ configurations prior to relaxation using DFT. The configurations are shown from (i) to (v) in energetic order, with (v) resulting in the lowest energy GB configuration after the relaxation. Both DFT and EAM predict the initial configurations of the two vacancies sitting at a 1NN distance 2.6 Å apart, to be unfavorable. DFT identifies two vacancies initially separated by 3.7 Å as the most favorable initial V${}_{2}$ configuration (see Figure 5(av)). Hence, di-vacancies tend to dissociate close to this GB. Such behavior has previously only been identified for larger vacancy clusters where nanovoids were found to dissociate when interacting with migrating GBs [82].

Figure 5b,c include the lowest energy fully relaxed GB configuration with a di-vacancy using EAM and DFT, respectively. The incorporation of a di-vacancy into the GB causes significant relaxation effects that facilitate the formation of SIAs. In the case of EAM, (Figure 5b), the geometry optimization of the periodically translated GB with di-vacancy results in the shift of the main symmetry axis of the GB. Furthermore, Σ5 deltoids are transformed into filled deltoids. Similar structural changes have been reported in ref. [83]. On the other hand, in the DFT relaxed GB configuration with a di-vacancy, seen in Figure 5c, the initial Σ5 deltoids are transformed into a split-deltoid configuration through the formation of SIAs that sit within the excess free volume in the GB area. We note that significant differences in the relaxed GB structures are seen in Figure 5b,c. Like in the bulk, a considerably higher number of SIAs are formed within the GBs when using EAM (Figure 5b) compared to DFT (Figure 5c). The latter illustrates that EAM could not capture the same relaxation effects induced by the di-vacancy when compared with DFT. This stems from the trend noted above whereby EAM simulations favor the SIAs formation in Cu structures containing lower-coordinated sites, such as larger vacancy clusters and GBs with di-vacancies.

Finally, we employed NEB calculations to investigate the di-vacancy absorption into the GB. Figure 6 shows the V${}_{2}$ migration path and energy profile obtained using NEB with five images. The initial configuration corresponds to a di-vacancy in the bulk but within the absorption region of 6.9 Å defined in Figure 4c. The final configuration is the lowest energy configuration of a di-vacancy inside the GB seen in Figure 5(av). The di-vacancy configurations during the absorption process are shown in Figure 6b in the same order as in Figure 6a. DFT and EAM predict the lower than 0.05 eV difference between configurations corresponding to images 1 and 2. The absorption of the di-vacancy by GB results in the 2.0 eV gain in energy caused by the strong structural relaxation. The high 2.0 eV reverse barrier illustrates the strong binding energy of di-vacancy to GB.

This strong lattice relaxation prompted us to test how the size of the periodic cell affects the amount of energy gained when a di-vacancy moves toward the GB. A total of two simulation cell sizes are tested with DFT and four with EAM. The largest GB cell used for DFT contains 304 atoms, whereas, for EAM, the largest cell contains 606 atoms. As seen in Figure 6a, using both DFT and EAM, 304-atom or larger cells are required to achieve converged energy values. However, for this cell size, the barriers computed with both methods are in good agreement.

## 5. Conclusions

In this study, we have investigated the structure, migration paths and barriers of vacancy clusters in the bulk of fcc Cu and near the Σ5 twin grain boundary. In the bulk, several most stable cluster structures up to six vacancies have been identified. The most significant interaction between vacancies takes place within one lattice parameter distance. As expected, the DFT results in the bulk show that formation and cluster binding energies per vacancy decrease and increase, respectively, and the addition of a further vacancy becomes more favorable than the decomposition of the cluster as the number of vacancies increases.

Using both DFT and EAM, the migration paths and barriers for clusters of up to three vacancies were computed. To identify the elementary diffusion mechanism and calculate the migration barriers of larger clusters, DFT and EAM were used with ARTn. In the bulk, di-vacancies exhibited the lowest migration barrier amongst the examined clusters, which confirms previous studies in fcc metals. V${}_{4}$ migration was found to follow the same diffusion mechanism as that of di-vacancies. V${}_{4}$, V${}_{5}$, and V${}_{6}$ clusters have considerably higher migration barriers compared to V${}_{2}$.

The diffusion barrier of a di-vacancy was further reduced in the vicinity of the Σ5 twin grain boundary. The absorption of V${}_{2}$ by the GB induced the formation of SIAs within the free volume formed by the GB. In addition, the GB structure was found to undergo significant changes from the initial deltoid-like structure due to the incorporation of di-vacancy. Our results also showed that di-vacancies prefer to dissociate inside the GB, a behavior that was previously shown only for larger vacancy clusters. Hence, these results indicate that for the initiation of voids in these regions, the presence of other factors may warrant consideration, such as additional point defects or impurities.

Finally, since a number of previous studies employed forcefield methods to model vacancy clusters in fcc metals, we evaluated the accuracy of EAM compared to DFT calculations. Both EAM and DFT demonstrate that clusters prefer to form closed-packed structures. Excluding V${}_{5}$, the same lowest energy cluster configurations were identified using DFT and EAM. The EAM potential [40] proved to be reliable for describing the binding and formation energies of vacancies in the bulk of Cu. The EAM results are in agreement with DFT regarding the lowest energy structures of V${}_{3}$, V${}_{4}$, and V${}_{6}$ clusters within the bulk and the description of the behavior of vacancies within the GB and the vacancy-induced relaxation effects. Furthermore, the EAM results are in good agreement with DFT in predicting the energy gain due to the migration of a di-vacancy in the GB, provided a sufficiently large periodic cell is employed. However, when compared with DFT, EAM did not capture the relaxation effects induced in the GB due to the presence of a di-vacancy. More significant discrepancies between the DFT and EAM results were observed in cases where structures with a high number of low-coordinated sites were considered, such as in penta-vacancies and grain boundaries.

These results are not only useful for our understanding of the behavior of vacancy clusters in fcc metals, but also provide a direct comparison between EAM and DFT methods that can be used as a reference for future studies. Our approach can be employed to provide a better understanding of structure and diffusion properties of more complex structures, such as complexes of vacancies with metallic and non-metallic impurities.

## Figures and Tables

**Figure 1 nanomaterials-13-01464-f001:**
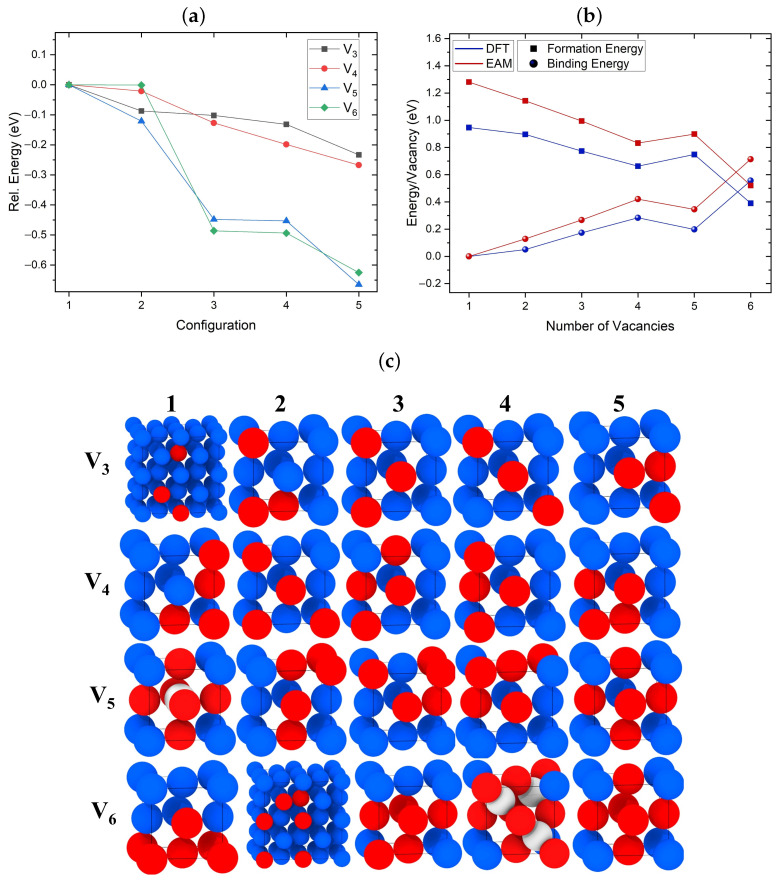
(**a**) Relative energies of the lowest energy configurations of V${}_{3}$, V${}_{4}$, V${}_{5}$, and V${}_{6}$ clusters identified using the SOD code and optimized with DFT. The highest energy configuration for each cluster is used as a reference. (**b**) Binding and formation energies per added vacancy calculated using EAM and DFT. For each cluster, the lowest energy configuration identified through DFT is considered. (**c**) Schematic illustrations of the lowest energy cluster configurations for V${}_{3}$–V${}_{6}$ clusters optimized using DFT. The configurations 1–5 are shown in the same energetic order as in (**a**). Cu atoms are blue, vacancies are red, whereas SIAs are shown in white.

**Figure 2 nanomaterials-13-01464-f002:**
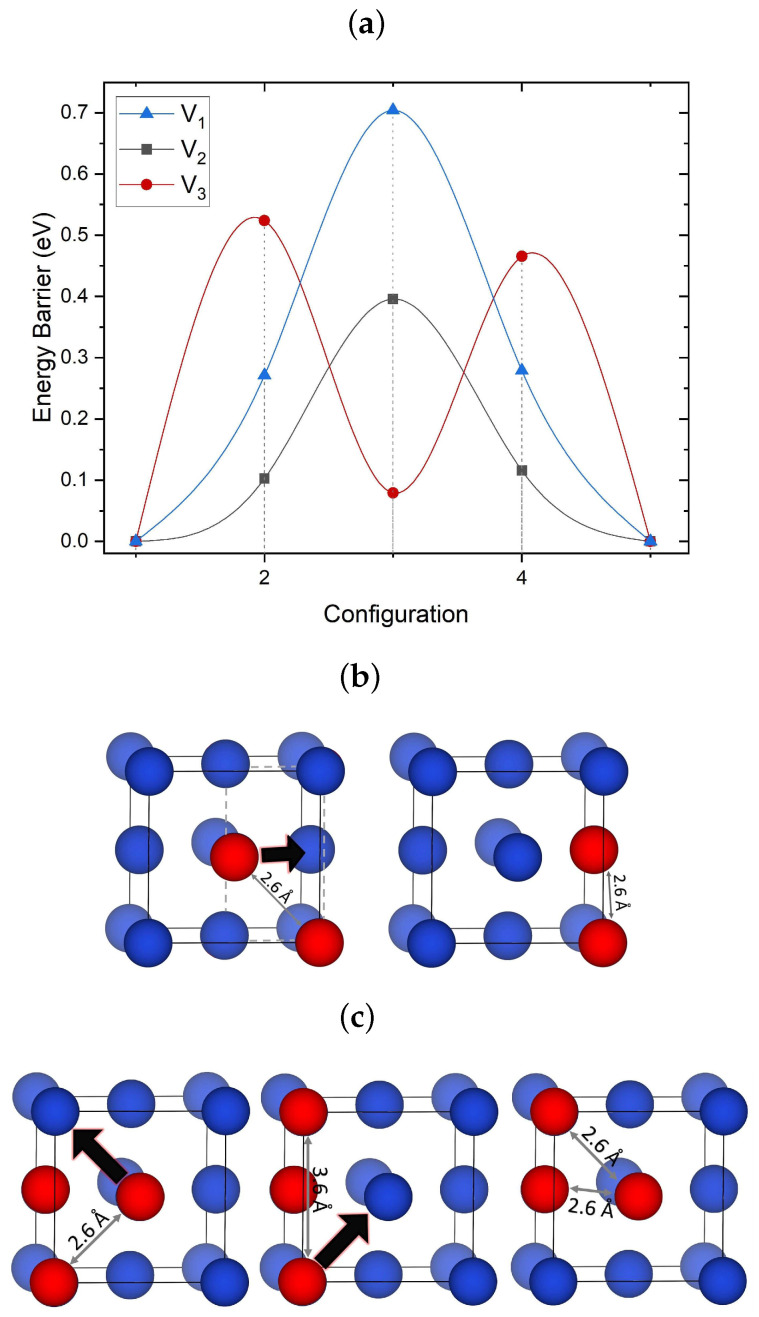
(**a**) Migration barriers of V${}_{1}$, V${}_{2}$ and V${}_{3}$ calculated using DFT and CI-NEB. The most favorable migration path is included for each cluster. (**b**) Schematics of the most favorable migration paths of V${}_{2}$ and (**c**) V${}_{3}$ identified using DFT/NEB. Black arrows depict the direction of vacancy displacement during cluster migration. Both clusters diffuse via a single vacancy hop mechanism. Cu atoms are shown in blue whereas vacancies are shown in red.

**Figure 3 nanomaterials-13-01464-f003:**
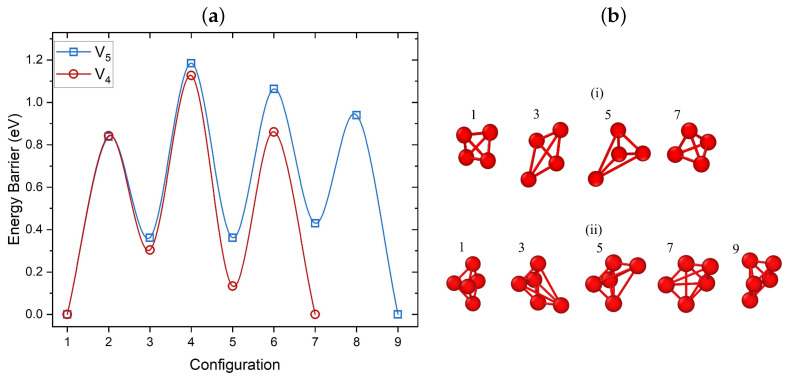
(**a**) Barriers for preferred paths of V${}_{4}$ and V${}_{5}$ cluster migration calculated using DFT and ARTn. For both V${}_{4}$ and V${}_{5}$, the initial and final configurations of the migration path correspond to the already identified lowest energy structures. (**b**) Local minima configurations for the migration of (**i**) V${}_{4}$ and (**ii**) V${}_{5}$, with the same order as seen in (**a**). Cu vacancies are shown in red. Both clusters migrate via a single vacancy hop mechanism.

**Figure 4 nanomaterials-13-01464-f004:**
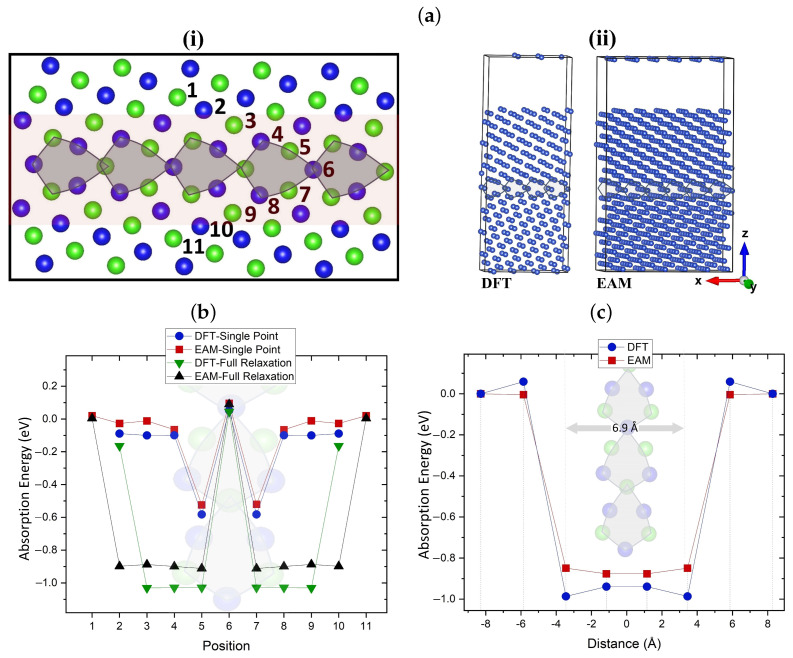
(**a**) (**i**) [100](210) Σ5 twin grain boundary used as an example of a vacancy sink. Blue and green colors show Cu atoms that sit in different planes. Numbers indicate the examined positions of a vacancy in the GB. The red-colored region illustrates the interaction range between GBs and vacancies. (**ii**) 304-atom (left) and 912-atom (right) Σ5 GB simulations cells used for DFT and EAM, respectively. (**b**) Single point (prior to relaxation) and fully relaxed absorption energies of mono-vacancies at different sites in the grain boundaries. (**c**) Absorption energy calculations of fully relaxed mono-vacancies at varying distances from the GB. In (**b**,**c**), the zero energy corresponds to vacancy in the bulk. The interaction range between vacancies and GB (absorption region) is approximately 6.9 Å wide. In both (**b**,**c**) plots, inset images of the grain boundaries are included.

**Figure 5 nanomaterials-13-01464-f005:**
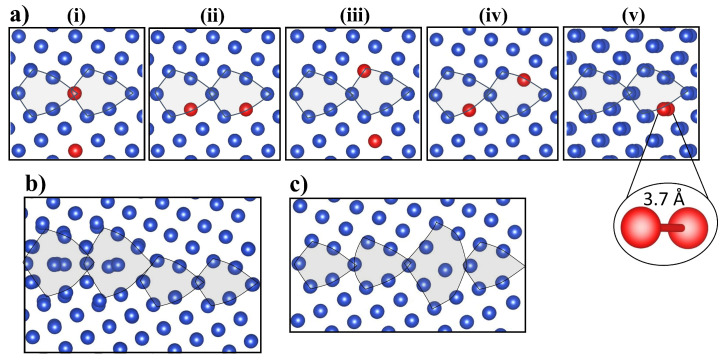
(**a**) (**i**–**v**) Lowest energy initial configurations of di-vacancies prior to relaxation. Configurations are shown from (**i**) to (**v**) in energetic order, with (**v**) resulting in the lowest energy GB configuration after relaxation. Both DFT and EAM identified the configuration shown in image (**v**), where two vacancies relax at a 3.7 Å distance, as the most favorable initial di-vacancy configuration. (**b**) The lowest-energy fully relaxed configuration with a di-vacancy absorbed by the grain boundary using EAM. The di-vacancy causes the shift of the main symmetry axis of GB. (**c**) The lowest-energy fully relaxed configuration of the same GB with absorbed di-vacancy using DFT. Cu atoms are shown in blue whereas vacancies are shown in red.

**Figure 6 nanomaterials-13-01464-f006:**
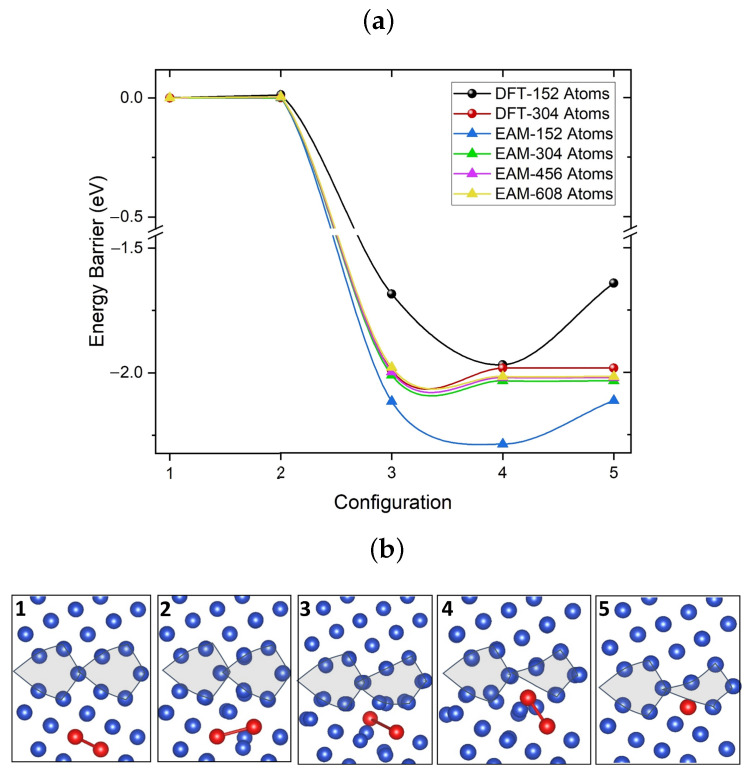
(**a**) Energy profiles for migration of a di-vacancy from the bulk towards the [100](210) Σ5 grain boundary axis calculated using both EAM and DFT with different cell sizes. (**b**) Illustrates the di-vacancy configurations along the migration path from the bulk into the GB. Configurations are shown in the same order as in (**a**). In the final configuration, the two vacancies are separated by 3.7 Å along the axis perpendicular to the figure plane.

**Table 1 nanomaterials-13-01464-t001:** Calculated migration barriers of V${}_{1}$–V${}_{6}$ clusters.

Cluster	Barrier (eV)	Method
V${}_{1}$	0.65	DFT/NEB
V${}_{2}$	0.40	DFT/NEB
V${}_{3}$	0.52	DFT/NEB
V${}_{4}$	0.84	DFT/EAM/ARTn
V${}_{5}$	0.84	DFT/EAM/ARTn
V${}_{6}$	0.96	EAM/ARTn

## Data Availability

The data presented in this study are available on request from the corresponding author.

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
