# Peer review of "Structure and Migration Mechanisms of Small Vacancy Clusters in Cu: A Combined EAM and DFT Study"

_nanomaterials, 2023, doi:10.3390/nano13091464_

Round 1

Reviewer 1 Report

Voids in face-centered cubic (fcc) metals, which are commonly assumed to form by the aggregation of vacancies, would lead to severe degradation effects. However, the mechanisms of vacancy clustering and diffusion are not fully understood at present. Herein, Fotopoulos et al. used the embedded atom method (EAM) potentials and density functional theory (DFT) to uncover a detailed insight into the structures and formation energies of primary vacancy clusters, mechanisms and barriers for their migration in bulk copper, and how these properties are affected at simple grain boundaries. They found that Migration of vacancy clusters occurs via hops of individual constituent vacancies with di-vacancies having a significantly smaller migration barrier than mono-vacancies and other clusters. This barrier is further reduced when di-vacancies interact with grain boundaries. The manuscript provides further insights into the structural evolution of metal films under thermal and mechanical stress. The reported results are of wide interest, of fine significance and of good impact on the researchers working in the fields of materials mechanics, physics, and physical chemistry. Nevertheless, there are some technical issues that required to be clarified. Therefore, a major revision should be made before the manuscript could be accepted.

The following points are my concerns that could help the authors to improve the clarity, significance and readability of the manuscript and thus they should be well-addressed and included in a revised version of the manuscript.

(1) Lines 5-8, it is not necessary to capitalize the first word of the full name of an abbreviation. Therefore, the sentence " The calculations were carried out using Embedded Atom Method (EAM) potentials and Density Functional Theory (DFT) and employed the Site-Occupation Disorder code (SOD), the Activation Relaxation Technique nouveau (ARTn) and the Knowledge Led Master Code (KLMC)." could be revised to " The calculations were carried out using the embedded atom method (EAM) potentials and density functional theory (DFT) and by means of the site-occupation disorder code (SOD), the activation relaxation technique nouveau (ARTn) and the knowledge led master code (KLMC)".

(2) It is well-known that the Generalized Gradient Approximations (GGAs) such as PBE functional and even hybrid functionals such as B3LYP functional substantially underestimate the van der Waals dispersive interactions and dispersion corrections should be used in the energy calculations [see: Yu, Y.-X. High storage capacity and small volume change of potassium-intercalation into novel vanadium oxychalcogenide monolayers V2S2O, V2Se2O and V2Te2O: An ab initio DFT investigation. Applied Surface Science 2021, 546, 149062. https://doi.org/10.1016/j.apsusc.2021.149062; Yu, B. J.; Piao, X. L.; Ren, H. Towards adsorptive enrichment of flavonoids from honey using h-BN monolayer. ChemPhysChem 2022, 23, e202100828. https://doi.org/10.1002/cphc.202100828]. Have the authors included the dispersion corrections in the system energy calculations?. If the authors did not consider the dispersion corrections, they should pointed out that in general case the dispersion corrections should be included in the PBE energy calculations as has done in above literature but why they did not consider it. If the authors did consider the dispersion corrections, please point out that they have considered the dispersion corrections in the PBE energy calculations just like that in above literature.

(3) The authors stated at Lines 26 and 27 that voids are commonly formed in fcc metals under high-stress conditions [7–9] leading to severe degradation. It should be pointed out that Zhang et al. found that a higher stress triaxiality facilitates the formation and growth of micro-voids, which leads to a decrement of strain at failure for rolled Ti6Al4V titanium alloy [Zhang, H.; Gao, T.; Chen, J.; Li, X. P.; Song, H. P.; Huang, G. Y. Investigation on strain hardening and failure in notched tension specimens of cold rolled Ti6Al4V titanium alloy. Materials 2022, 15, 3429. https://doi.org/ 10.3390/ma15103429]. This paper can be used as evidence for the statement.

(4) The format of the article title of Ref. [1] should be revised to " A continuum model for void nucleation by inclusion debonding ". The same issue also exists in Refs. [5], [6], [36], [44], [52], and [68].

(5) In Figures 2 and 5, the physical meaning of blue and red balls should be explained in the figure or in the figure caption for clarity.

(6) The unit of the values of cluster barriers in Table 1 should be added.

Author Response

First referee

We are grateful to the referee for their positive and insightful review of our manuscript. We address their comments in detail below.

  • Lines 5-8, it is not necessary to capitalize the first word of the full name of an abbreviation. Therefore, the sentence " The calculations were carried out using Embedded Atom Method (EAM) potentials and Density Functional Theory (DFT) and employed the Site-Occupation Disorder code (SOD), the Activation Relaxation Technique nouveau (ARTn) and the Knowledge Led Master Code (KLMC)." could be revised to " The calculations were carried out using the embedded atom method (EAM) potentials and density functional theory (DFT) and by means of the site-occupation disorder code (SOD), the activation relaxation technique nouveau (ARTn) and the knowledge led master code (KLMC)".

All of the acronyms' complete names (DFT, KLMC, EAM, SOD, ARTn, GGA, GPW) are no longer capitalized. Additionally, we made sure to fix the problem in the revised manuscript's abbreviations section.

  • It is well-known that the Generalized Gradient Approximations (GGAs) such as PBE functional and even hybrid functionals such as B3LYP functional substantially underestimate the van der Waals dispersive interactions and dispersion corrections should be used in the energy calculations [see: Yu, Y.-X. High storage capacity and small volume change of potassium-intercalation into novel vanadium oxychalcogenide monolayers V2S2O, V2Se2O and V2Te2O: An ab initio DFT investigation. Applied Surface Science 2021, 546, 149062. https://doi.org/10.1016/j.apsusc.2021.149062; Yu, B. J.; Piao, X. L.; Ren, H. Towards adsorptive enrichment of flavonoids from honey using h-BN monolayer. ChemPhysChem 2022, 23, e202100828. https://doi.org/10.1002/cphc.202100828]. Have the authors included the dispersion corrections in the system energy calculations?. If the authors did not consider the dispersion corrections, they should pointed out that in general case the dispersion corrections should be included in the PBE energy calculations as has done in above literature but why they did not consider it. If the authors did consider the dispersion corrections, please point out that they have considered the dispersion corrections in the PBE energy calculations just like that in above literature.

We agree that dispersion corrections are important for describing the properties of layered materials cited by the referee, and have been concerned about including them also into our bulk fcc Cu calculations. However, previous theoretical studies (Philos. Mag. (2014) 2;94(31):3522-48, Acta Materialia (2018) 148(4), Metals (2023) 13 (2), 346) clearly suggest that including dispersion corrections in bulk metal calculations is not required. As mentioned in the revised manuscript, the van der Waals interactions are negligible in pure metallic systems like Cu.

  • The authors stated at Lines 26 and 27 that voids are commonly formed in fcc metals under high-stress conditions [7–9] leading to severe degradation. It should be pointed out that Zhang et al. found that a higher stress triaxiality facilitates the formation and growth of micro-voids, which leads to a decrement of strain at failure for rolled Ti6Al4V titanium alloy [Zhang, H.; Gao, T.; Chen, J.; Li, X. P.; Song, H. P.; Huang, G. Y. Investigation on strain hardening and failure in notched tension specimens of cold rolled Ti6Al4V titanium alloy. Materials 2022, 15, 3429. https://doi.org/ 10.3390/ma15103429]. This paper can be used as evidence for the statement.

We thank the referee for bringing these papers to our attention. The mentioned statement is now included in the introduction of the revised manuscript. The paper by Zhang et al. (https://doi.org/ 10.3390/ma15103429) is cited as a justification.

  • The format of the article title of Ref. [1] should be revised to " A continuum model for void nucleation by inclusion debonding ". The same issue also exists in Refs. [5], [6], [36], [44], [52], and [68].

All of the mentioned references have now been revised and all issues have been resolved. Also, we made sure any additional publications in the revised manuscript adhered to the suggested format.

  • In Figures 2 and 5, the physical meaning of blue and red balls should be explained in the figure or in the figure caption for clarity.

We thank the referee  for pointing this out. To address the issue, we included a more thorough explanation in all of the relevant figure captions, clearly mentioning the meaning for each color.

  • The unit of the values of cluster barriers in Table 1 should be added.

Thanks for this observation. The diffusion barriers' units (eV) are now included in Table 1.

Reviewer 2 Report

1. The authors are suggested to provide more details introduction section:

a> For example, the application of copper should be discussed in more detail and a short comparison with other competitors should be provided for similar applications. Please see:  ACS Applied Energy Materials 4 (12), 14043-14058, ACS omega 4 (1), 971-982, Sustainable Energy & Fuels 3 (7), 1668-1681

b> The authors wrote: "Since in metals like Cu and Al [12,18,24], such small clusters are not easily identified 38 experimentally, theoretical modelling is often used to determine their characteristics." please briefly introduce the technical issue for such bottleneck and also provide the details of any published study.

2. Please mention the meaning of red, white and blue balls in Figure 1. 

3. Is there any difference between migration behaviours for nano and bulk systems?

4. Please mention the meaning of different color balls in Figure 2-3-4. 

Please check the text. there are some texts with hard understanding.

Author Response

Second referee

We thank the second referee for their valuable comments, which we address in detail below.

  • The authors are suggested to provide more details introduction section:

  • For example, the application of copper should be discussed in more detail and a short comparison with other competitors should be provided for similar applications. Please see: ACS Applied Energy Materials 4 (12), 14043-14058, ACS omega 4 (1), 971-982, Sustainable Energy & Fuels 3 (7), 1668-1681

We thank the referee for bringing these relevant publications to our attention. We agree that the introduction could highlight more clearly the wide range of applications for Cu. In response to this comment, we have updated the introduction of the revised manuscript to include citations for the suggested papers (ACS omega 4 (1), 971-982, Sustainable Energy & Fuels 3 (7), 1668-1681) along with another very informative publication on Cu interconnects (JOM (2001) 53, 43–48, https://doi.org/10.1007/s11837-001-0103-y).

  • The authors wrote: "Since in metals like Cu and Al [12,18,24], such small clusters are not easily identified 38 experimentally, theoretical modelling is often used to determine their characteristics." please briefly introduce the technical issue for such bottleneck and also provide the details of any published study.

We agree that this point is not fully addressed in the original manuscript. In response, we have included more discussion of previous experimental research of vacancies in metals as well as the role of theoretical simulations. As mentioned in the revised manuscript, although methods like positron annihilation spectroscopy (PES) can be used to investigate vacancies in metals, theoretical simulations are still needed to interpret the experimental data. Also, due to the limited resolution of electron microscopes and radiation damage, it is still challenging to observe single defects like vacancies experimentally using high resolution TEM (see e.g. Scientific Reports 4, 3683 (2014)). Please also refer to the papers by Yang et al. (Journal of Nuclear, 571, 154019 (2022)) and Eyre (J. Phys. F: Met. Phys. 3 422 (1973)). Therefore, theoretical simulations play an important role in predicting the structural properties and migration mechanisms of small vacancy clusters in metals. Finally, we added a relevant publication on experimental studies in fcc metals (Materials Science and Engineering, 350, 15, 95-101 (2003)).

The referee also made the following points:

  • Please mention the meaning of red, white and blue balls in Figure 1.

A similar comment was made by the first referee (comment no.5). The issue, as already mentioned, has now been resolved.

  • Is there any difference between migration behaviours for nano and bulk systems?

We assume that the reviewer is referring to the migration of vacancies and vacancy clusters discussed here. This is an interesting and complex question and our results provide only some hints towards a full answer. In particular, our results demonstrate that migration of Cu vacancies towards grain boundaries is affected from the distance of about 3.5 Å from the GB plain. Similar effects are expected near surfaces and facets on nanoclusters. This has been demonstrated in the detailed study by C. van der Walt et al.  “A study of diffusion, atom migration and segregation in Cu and Ag alloy bulk- and nanocrystals”, AIP Advances 7, 055102 (2017). The migration barrier depends on the proximity to the surface and direction of travel. We are unaware of similar studies for vacancy cluster diffusion in pure metal nanoclusters.

  • Please mention the meaning of different color balls in Figure 2-3-4.

As already mentioned in comment no.2, the issue has now been resolved.

Round 2

Reviewer 1 Report

The authors have responded to all the comments and suggestions raised by the Reviewers. The authors have revised the representation of an abbreviation and All of the acronyms' complete names (DFT, KLMC, EAM, SOD, ARTn, GGA, GPW) are no longer capitalized in the revised manuscript. Additionally, the authors have made sure to fix the problem in the revised manuscript's abbreviations section. The authors have included a more thorough explanation in all of the relevant figure captions, clearly mentioning the meaning for each color in the revised version of the manuscript. The revised version of the manuscript is approaching to be accepted. However, one of the important issue in computational method has not described clearly.

The authors have correctly answered the question (2) raised by the Reviewer but they were not clear stated in the revised manuscript because all in all, the dispersion correction should be included in the energy calculations. According to the authors' respond, the following sentences should be added to "2.1.2. DFT Calculations":

"Although the dispersion correction is a considerable part of energy in common adsorption systems [Yu, Y.-X. High storage capacity and small volume change of potassium-intercalation into novel vanadium oxychalcogenide monolayers V2S2O, V2Se2O and V2Te2O: An ab initio DFT investigation. Applied Surface Science 2021, 546, 149062. Yu, B. J.; Piao, X. L.; Ren, H. Towards adsorptive enrichment of flavonoids from honey using h-BN monolayer. ChemPhysChem 2022, 23, e202100828.], but previous investigations [Ganchenkova, M.; Yagodzinskyy, Y.; Borodin, V.; Hänninen, H. Effects of hydrogen and impurities on void nucleation in copper: simulation point of view. Philosophical Magazine 2014, 94, 3522–3548] identified that it is negligible for metal systems [Philos. Mag. (2014) 2;94(31):3522-48, Acta Materialia (2018) 148(4), Metals (2023) 13 (2), 346) ]. Therefore in accordance with previous DFT simulations on vacancies in transition metals, the dispersion corrections have not been used in this work".

The addition of above sentences and related references would helpful for readers to understand why the authors neglect dispersion correction which is very important for common adsorption systems.

The quality of English language is fair in the revised manuscript.
